# The Pedagogical Use of Gamification in English Vocabulary Training and Learning in Higher Education

Benjamin Panmei [1] and Budi Waluyo [2,*]

1   Language and Culture for Business Department, Bangkok University (Bangkok University International), Bangkok 12120, Thailand
2   Languages Department, Walailak University, Tha Sala 80160, Thailand
*   Correspondence: budi.business.waluyo@gmail.com

**Abstract:** One of the persistent challenges in vocabulary teaching is that EFL students must learn a certain number of words to operate in English, yet class time is limited. While the usage of gamification applications may alleviate some of these issues, research on the usefulness of gamification apps and their potential to assist student vocabulary learning outside of the classroom is currently limited. This study used a quasi-experimental method to investigate the effects of gamified vocabulary learning using an application called *Quizizz*. It compared the learning outcomes of experimental and control groups, as well as the usefulness of gamification in boosting learner autonomy in vocabulary learning. Multiple independent *t*-tests revealed non-significant differences between students' overall vocabulary test results in both groups ($t(2, 98) = 1.920$, $p = 0.06$) with a modest effect size (Cohen's d = 0.3). Significant differences were noticed in the results of vocabulary tests 2 ($t(2, 98) = 3.229$, $p = 0.002$) and 4 ($t(2, 98) = 3.465$, $p = 0.001$), where the students in the experimental groups achieved higher scores ($M = 13.63$ and $M = 12.60$), yet no significant differences were noted in the results of vocabulary tests 1 and 2. Significant changes occurred in students' perceptions of the application of gamification as a means for facilitating training ($t(1, 49) = 2.269$, $p = 0.03$) with a modest effect size (Cohen's d = 0.2). Nonetheless, their perceptions of the use of gamification for enhancing learner autonomy in vocabulary learning did not significantly change before and after the interventions ($t(1, 49) = 1.652$, $p = 1.05$) with a very small effect size (Cohen's d = 0.2). For pedagogy and research, the findings enhance our understanding that gamification apps can be integrated into vocabulary learning to facilitate vocabulary study and foster the growth of learner autonomy, yet utilizing a non-specifically built app for vocabulary learning may not produce better results than those who do not receive gamification support, even if learning outcomes remain good. More research exploring the impacts of integrating a non-specifically built app into English language teaching and learning is needed.

**Keywords:** learner autonomy; gamification; Quizizz; training; vocabulary





## 1. Introduction

Gamification has gradually altered the landscape of English vocabulary learning. Since 2016, the number of research publications in the area has tripled [1]. Based on a systematic review of studies published from 2012 to 2021, the increased interest in the use of gamification in English classes is primarily driven by the ability to use mobile apps to aid in the creation of mobile learning environments, which has been widely implemented to improve students' vocabulary learning [2]. Several empirical studies have shown that incorporating gamification into English vocabulary classes can transform the learning experience from being tedious and boring, emphasizing memorization and repetition, to becoming playful [3], motivating, and engaging [4,5], while still remaining within comprehensive vocabulary instruction and promoting other key elements in learning, such as problem-solving and collaboration [6]. The positive effects on learning outcomes have

been observed not only among high proficiency students, but also among students with low English proficiency levels [7].

Recently, there have been an increasing number of studies examining the use of gamification as a means for facilitating vocabulary training and enhancing learner autonomy. One of the motivating factors is that a certain number of words must be learned to function in the English language [8], which cannot be regularly covered by teaching the target words during the class hour. Students who are learning English as a foreign language (EFL) must have the knowledge of between 2000 and 3000 words to participate in basic everyday conversations and read authentic English texts [9]. To perform more complicated oral discourses and comprehend more sophisticated levels of written texts, students need to know 8000–9000- and 5000–7000-word families, respectively [10,11]. In some countries, thresholds for vocabulary learning instructions have been implemented to ensure that students can acquire the number of required words for performing academic and communicative tasks in English. In Japan, for instance, students receive 800–1200 h of instruction to learn 2000–2300 words; in Indonesia, students receive 1200 h of instruction to learn 1200 words. In China, students majoring in English receive 1800–2400 h of instruction to learn 4000 words [12]. In Thailand, students are required to study 3000 English words over 216 instructional hours during their first two years of university education [4]. The policy on English teaching and learning might vary among Thai universities, depending on the university's policy. However, formal vocabulary learning is still searching for optimal instructional designs. The difficulties stem from the fact that class time is limited, teachers have a greater number of lessons to cover, and each word comes with its own set of complications [13].

On the other hand, gamification apps can give English teachers the opportunity to extend their vocabulary learning instructions to learning activities outside the classroom as well as the ability to monitor students' progress continuously. Studies that integrated gamification apps into students' outside-classroom vocabulary learning activities confirmed that the gamified learning encouraged learners to be agents of their own learning and stimulated interest-driven learning [14–16]. After reviewing twenty-one research publications on digital game-based vocabulary learning in SSCI indexed journals, Zou et al. [17] conclude, "(1) digital games promote effective vocabulary learning; (2) interactions in game environments are conducive to vocabulary learning; (3) game-embedded multimedia facilitates vocabulary learning; and (4) over-specified vocabulary information is better than isolated or minimally specified information." (p. 22–23).

English teachers at a university level have recognized and used some internet-based applications in their classroom teaching practices, including *Kahoot*, *Socrative*, *Google Form*, *QR code*, *Facebook*, *YouTube*, *Quizizz*, and *Quizlet* [18]. Among these mentioned applications are gamification tools that can facilitate online quizzes, including *Kahoot*, *Quizizz*, and *Quizlet*. The use of these three apps in English learning has opened the path to the integration of mobile learning into an EFL setting, which is also supported by the fact that all university students own smartphones and use them daily. English teachers have favorable attitudes towards the use of mobile apps in class [19] although they note the importance of having proper planning and a clear policy on the integration of technology into English teaching [20]. Of the available apps, *Quizlet* is one of the most frequently utilized apps in vocabulary learning due to its specific features aimed at vocabulary classes [7,21] even though it can also be used for teaching other English skills, such as Grammar and Speaking. A mobile app called *Phone Words* was recently incorporated by Chen et al. [22] into the vocabulary learning of Taiwanese EFL learners, and the researchers discovered significant improvements in the learning performance of vocabulary acquisition and vocabulary retention, as well as perceived higher effectiveness and satisfaction with the use of the app for facilitating English vocabulary learning. The number of studies investigating the integration of other gamification apps into vocabulary learning outside the classroom is still scarce [23,24]. Moreover, a review study by Xu et al. [25] highlights the lack of rigor and transparency in the majority of published research on the use of digital game-based technol-

ogy for English language learning. Several apps, including "Duolingo: Learn Languages Free" and "Learn Languages-busuu" [26] and an in-house app called *Excel@EnglishPolyU* developed by a university in Hong Kong for learning business vocabulary [27], have been specifically created for language learning, especially vocabulary learning. However, popular apps, like *Kahoot!, Socrative,* etc., are not specifically designed for English teaching and learning but have been used by teachers [18,28]. Hence, the essence of exploring the effectiveness of gamified vocabulary learning lies in its practical implementation and the selected app.

## 2. Literature Review

### 2.1. Gamification as a Means for Facilitating Vocabulary Training

The idea of providing some materials to be used for learner training is not entirely novel. In the 1980s, early scholars argued that successful learners' strategies could be codified and taught to low-performing learners, resulting in an increase in learning efficiency. Implementation was suggested to be effective through the facilitation of learner training in target language features, as poor language learners could apply the strategies multiple times while completing lesson exercises [29,30]. This resulted in the development of textbooks for student training in the English Language Teaching (ELT) profession [31]. Numerous learner training exercises had been incorporated into the course books, and teachers had begun to incorporate additional Supplementary Materials for student practice in class [32]. Teachers were recommended to devote 20% or more of class time on learner training [31]. Nonetheless, there was some debate over whether dedicating a portion of class time to learner training was successful and worthwhile, given that teachers were also obligated to present other learning materials with a variety of target language skills during class hours [33]. At this point, early research considers learner training as a pedagogical strategy that uses additional Supplementary Materials for students to practice either within or outside of the classroom. Meanwhile, the current study seeks to broaden the educational method by incorporating a gamification app into students' vocabulary learning outside of the classroom. The underlying premise is the same, which is to provide students with supplementary vocabulary learning materials as part of their preparation for in-class vocabulary tests.

Over the last decade, several empirical studies have been conducted to investigate the effectiveness of using gamification apps to aid learner vocabulary training outside of the classroom. Waluyo and Bakoko (2021) [34], for example, investigated the incorporation of gamification into vocabulary learning classes supported using vocabulary lists in a university-level English course in Thailand. For ten weeks, they taught 500 academic English words at the A2-B2 levels of the Common European Framework of Reference for Languages (CEFR). The words were divided into ten vocabulary lists, each with 50 words. Students were assigned to study each vocabulary list at home by completing word meanings and sentence samples, and they were given weekly vocabulary tests in class. The study used two cycles: learners learned the words without the use of gamification (Cycle 1) and with the use of gamification (Cycle 2). The paired-sample *t*-test results showed a significant difference in learners' scores before and after gamification with a medium effect size (Cohen's d = 0.57). Learners reported moderately positive effects on their autonomy in vocabulary learning and on facilitating vocabulary training.

Similarly, in the USA, Dreyer [15] used a gamification app to mediate vocabulary classes for high school students. His research created *Quizlet* vocabulary sets for students to study at home once a week, followed by weekly vocabulary tests in class. Through a teacher's account, the teacher was able to monitor students' *Quizlet* activities. In summary, the study discovered that those who used *Quizlet* frequently for vocabulary learning and training at home outperformed those who used it less frequently in vocabulary tests. Providing *Quizlet* for students to study before coming to class can improve their participation in class. In another study in the Netherlands, Runhaar et al. [35] applied an experimental research design to provide learner vocabulary training in a reading class using a gamifica-

tion app. The intervention group received home-learning assistance from *Quizlet*, whereas the control group did not. The results showed that students in the intervention group spent less time reading on vocabulary and general language questions than those in the control group; they also became more active and engaged in reading class activities.

Cunningham [36] emphasizes, based on his experience using gamification in various EFL classes in the United States, ranging from community college to university levels, that a gamification app can serve as a place for vocabulary training for students, thereby accelerating the growth of autonomy in vocabulary learning. Prior to the last decade, implementing focused vocabulary-training programs was difficult because teachers had to create, manage, and disseminate vocabulary sets on their own. Nonetheless, with the advent of online apps that can support various training approaches in the form of gamified vocabulary exercises, implementing an online vocabulary training program outside of the classroom is no longer a problem. It enables the teacher to address issues such as the limited class time and the need to cover a certain number of English words in each period [37].

Studies on the pedagogical use of gamification apps for facilitating vocabulary training are still in their early stages, and empirical evidence is sparse. Nonetheless, this research topic can be linked to the usage of gamification apps for assisting students' outside-of-the-classroom online learning or asynchronous learning for other disciplines, but the number is similarly limited. For instance, Figueroa-Caas and Sancho-Vinuesa [38] incorporated gamified online quizzes to help students practice their mathematics homework in preparation for in-class assessments at the Open University of Catalonia, Spain. They discovered a link between learning achievement as evaluated by final exam marks and participation in online quizzes, as well as the learning gained from completing these quizzes. Lee-Thomas et al. [39] utilized Blackboard to integrate online gamified quizzes as graded coursework for Mechanical Engineering students at a university in the southeast of the USA. They discovered that time spent, number of attempts, and quiz scores all had a favorable effect on student performance. Kimbrel and Gantner [40] integrated online quizzes into an internet platform for Instructor-Made Videos (IMVs) in an Educational Leadership Program for initial K12 leadership certification in Georgia, USA. The findings suggested that participants viewed quizzes as a useful addition to IMVs that kept all students accountable for watching the entire movie. Given the rise in the number of graduate courses offered online, this study demonstrates that combining Instructor-Made Videos with quizzes is an excellent technique for increasing student engagement and learning. All these studies indicate an increasing interest in using gamification apps to help students learn outside of the classroom, but there is still a lack of study on the subject.

Further, despite the promising findings of studies investigating the use of gamification to facilitate student vocabulary training outside the classroom, it is critical to note that most studies used the gamification app *Quizlet*. This app is based on the concept of vocabulary learning through flashcards. It extends the concept of learning to the mobile learning environment by including additional writing, spelling, and matching activities. In other words, the app is well-suited for integration into vocabulary classes. Nonetheless, one might reasonably wonder whether other gamification apps not specifically designed for vocabulary learning, such as *Kahoot!* and *Quizizz*, would have the same beneficial effect on student vocabulary learning. This is one of the knowledge gaps that the current study seeks to fill.

### 2.2. Gamification as a Means for Enhancing Learner Autonomy

Learner autonomy is defined as the capacity of a learner to assume responsibility for his or her own learning through the use of active, personally relevant strategies to accomplish learning objectives [41]. The significance is that once a learner takes ownership of his or her own learning, he or she will be able to set learning goals, manage appropriate learning methods, and become autonomously active in learning [42]. In other words, a high degree of learner autonomy enables a learner to perform autonomous learning activities and make independent decisions without the intervention of a teacher [43]. Particularly in the

context of vocabulary learning, ELT teachers believe that learner autonomy plays a critical role in stimulating intrinsic motivation and engagement during the process of vocabulary knowledge development [44], and EFL learners have a basic awareness of the importance of autonomous learning and the capacity to practice it, with further development dependent on how teachers design the learning environment for students [45].

When integrated into language learning instructions, research shows that modern technologies significantly increase language learners' autonomy [46]. Gamification apps are one of the technologies that enable teachers to foster learner autonomy outside of class hours. The majority of vocabulary learning that incorporates gamification to increase learner autonomy is accomplished by providing students with a set of words to explore at home via repetitive gamified exercises facilitated by the chosen gamification apps. Previous research has established that incorporating gamification improves learner autonomy in vocabulary learning [47,48], decreases forgetting rates, resulting in improved vocabulary knowledge retention [49], and improves vocabulary learning outcomes for low-proficiency students [7]. Students who receive gamification support are encouraged to manage their vocabulary practices at home independently. Simultaneously, the game elements integrated into a gamification application would increase students' enjoyment of learning, encouraging them to practice more with the app [50].

Prior research has also examined the use of gamification to increase students' learner autonomy in writing and reading classes that still include elements of vocabulary learning. For example, Sato et al. [51] integrated a gamification app into a writing class, assigning students in the experimental group to study English phrases created using the gamification app while students in the control group studied prepared English phrases using paper. The results of the writing tests revealed a significant difference in the students' use of English phrases in their writing, indicating that the experimental group students outperformed their counterparts. According to the survey results, students in the experimental group were more motivated to learn vocabulary and reported a greater degree of autonomy in their learning experience than those who used the traditional paper-based list. Previously, Sato et al. [52] integrated a multimedia application with features comparable to those found in current gamification applications into a reading class focused on vocabulary learning. The study discovered that using mobile devices to learn L2 vocabulary could improve vocabulary recall automation, freeing up cognitive resources for reading activities and thus successful L2 reading comprehension. However, the impact of gamified interventions on learning may be contingent on students' intrinsic and extrinsic motivations; thus, the teacher's role and guidance throughout the learning process in sustaining students' motivation cannot be overlooked [53].

### 2.3. Research on the Use of Quizizz in Vocabulary Learning

*Quizizz* is less popular than *Quizlet* in the gamification research literature for its application to vocabulary learning. Contrary to *Quizlet*, *Quizizz*'s design and features were not created specifically to aid in vocabulary learning. This is one of the reasons for the current study, which investigates whether gamification apps that are not specifically designed for vocabulary learning can still benefit student vocabulary learning in terms of facilitating learner training, enhancing learner autonomy, and improving learning outcomes, which may shed light on additional roles for gamification in vocabulary learning in the context of non-classroom learning. Within the small number of studies that have been conducted, the use of *Quizizz* in vocabulary classes has been found to be beneficial due to its effectiveness, feasibility, ease of use, and motivating nature for students [24]. Amalia [54] discovered, " . . . (that) the students strongly agreed that *Quizizz* has an attractive display which is interesting and fun, students can't cheat during the test, *Quizizz* creates a competitive atmosphere in the classroom, and *Quizizz* is better than the offline traditional test" (p. 1).

Katemba and Sinuhaji [23] applied the ESA (Engage, Study, Activate) method through *Quizizz* to enhance students' vocabulary knowledge. The study revealed that students in the experimental group achieved a higher level of proficiency and enjoyed vocabulary

learning more than students in the control group who did not receive *Quizizz* assistance. In comparison, Bal [55] examined the relationship between MALL and vocabulary learning in a 4-week study involving 60 students divided into experimental and control groups. The results indicated that the experimental group slightly outperformed the control group, though there was no statistically significant difference in their scores. According to Göksün and Gürsoy [56], the *Quizizz* application's limited visual feedback capacity, question progress at an individual pace, and technological issues may all be disadvantages of *Quizizz*. These contradictory findings underscore the importance of additional empirical research, to which the current study intends to contribute by expanding the investigation of the use of *Quizizz* for facilitating learner training and enhancing learner autonomy. In conclusion, while these three studies demonstrated the explicit educational application of Quizizz in vocabulary learning, empirical evidence remains sparse. There is a need to understand how the Quizizz app can be utilized outside of the classroom to aid in the development of learner autonomy in vocabulary learning, which has not been covered in detail in these three studies.

*2.4. The Study*

The present study builds upon the intention to examine the effects of gamified vocabulary learning through one of the gamification apps named *Quizizz* in a quasi-experimental research design. The study specifically explores students' perceptions of the use of a gamification app as a means for facilitating their vocabulary training outside the classroom and its effectiveness in enhancing learner autonomy in vocabulary learning. It also looks at how gamified vocabulary learning affects students' vocabulary learning outcomes. The following research questions are addressed:

1.   To what extent does the pedagogical application of gamification in this study improve students' vocabulary learning outcomes when compared to a control group studying vocabulary independently without gamification support?
2.   How do students perceive the pedagogical application of gamification as a means for facilitating vocabulary training and enhancing learner autonomy and learning outcomes?

## 3. Methods

*3.1. Research Design*

The present study examined the effects of gamified vocabulary learning when implemented to facilitate learner training and explored the effects on advancing learner autonomy and learning outcomes. To achieve such objectives, a quasi-experimental research design involving experimental and control groups was chosen. The design allows the addition of specific interventions to the experimental groups while keeping the control group with no special treatments; both groups are believed to share similar characteristics, thereby permitting the examination of the comparison of the outcomes at the end of the process [57]. In this study, the research design divided the students into two groups: experimental and control groups without random assignments. The experimental groups received the gamified learning intervention during their vocabulary learning while the other group did not. The control group studied the weekly vocabulary sets in an MS Word file, which they could print out. Survey questionnaires exploring students' perspectives regarding their learning experience were distributed before and after the learning process. The experimental group was the focus of this study, whereas the control group served as a comparison of the learning outcomes. Figure 1 below illustrates the research design.

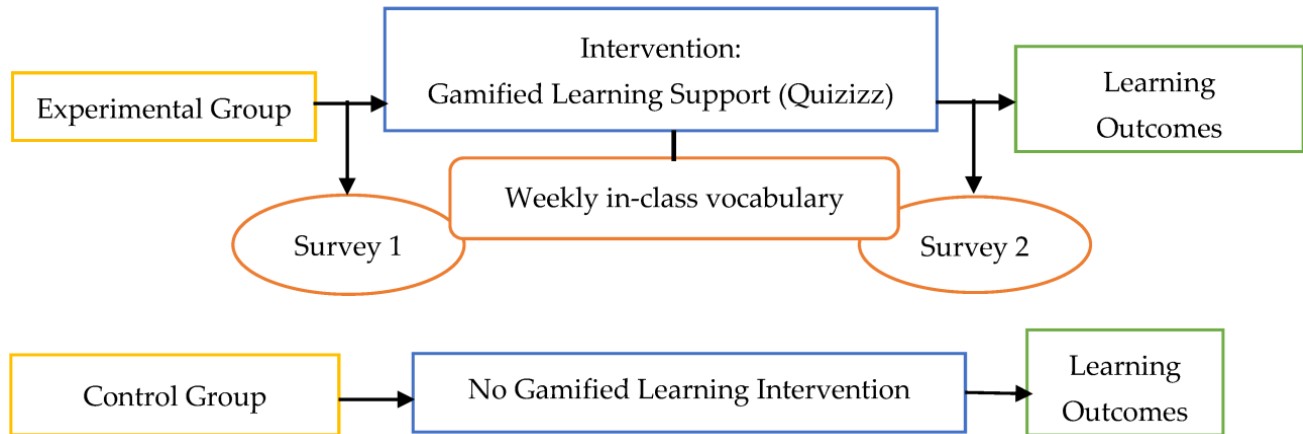

**Figure 1.** Illustration of the research design.

Ethical concerns. This study had been approved by the institutional review board of Walailak University, Thailand (WUEC-21-237-01). Participation was entirely voluntary and had no effect on the participants' grades or academic performance. The participants' personal information was kept private, and this study identified each student by identification numbers to which the researchers had access. The CITI Program in the United States and Walailak University in Thailand have certified the researchers for socially and behaviorally responsible research.

### 3.2. Context and Participants

The study was carried out at an autonomous university located in the southern region of Thailand. The participants were undergraduate students studying General English (GE) courses, i.e., GEN 61–121 English Communication Skills. After receiving approval from the university's Institutional Review Board, participants were recruited using a purposive sampling method [58] with three established criteria. The participants had to be 1) first-year undergraduate students, 2) enrolled in general English courses during the data collection, and 3) at the level of basic users (A1-A2) of English according to the Common European Framework of Reference (CEFR). Correspondingly, there were 100 participants, consisting of 57 females and 43 males. According to the results of the university placement test, the English proficiency levels of the students were at the level of basic users of English (A1-A2) based on the Common European Framework of Reference (CEFR). The participants were divided into two groups of an equal number. One of the researchers was in charge of instructing both the experimental and control groups. The teacher maintained objectivity by just following the course designs and learning instructions provided by the course coordinator and making no contributions. The other researcher was not teaching both groups and had contributed nothing to course development. We made every effort to avoid any research bias that may have occurred during the research period.

Experimental group. It consisted of 50 first-year undergraduate students (60% females and 40% males), aged 18–22 years old. The students had different academic majors, including Economics, Medical Technology, Hospitality Industry and Food Service, Food Science and Innovation, and Tourism and Hospitality. This experimental group completed the survey questionnaires distributed before and after the intervention was given. The gamification intervention was applied to this group following the research procedures.

Control group. It comprised 50 first-year undergraduate students (54% females and 46% males), aged 18–20 years old. The students majored in Medical Technology, Food Science and Innovation, and Liberal Arts. This group received no gamification intervention and served as a comparison group concerning the learning outcomes.

*3.3. Instruments*

The data were gathered using three different instruments. The first instrument was a survey questionnaire that was distributed to the experimental group via *Google Form* before and after the vocabulary learning. The students had past experience learning using the gamification app (Quizizz.com); therefore, the survey prior to the intervention was designed to obtain data on their thoughts of whether the app would be good for their vocabulary learning outside of the classroom. Meanwhile, the post-survey would reveal whether or not their perceptions had shifted as a result of the study's intervention.

A set of Likert-scale questionnaires was developed to collect students' perceptions of their experience with *Quizizz* in vocabulary learning. The impact of *Quizizz* on the development of autonomy was measured in two ways: by students' perceptions of the use of *Quizizz* as a place for learner vocabulary training, and by the impact of *Quizizz* on the development of learner vocabulary training. The former had eight items, such as "*Quizizz* from teacher really helped me learn the vocabulary sets more and better.", "*Quizizz* from teacher really facilitated my vocabulary learning through practices.", and "*Quizizz* from teacher enabled me to practice on vocabulary exercises more." The items were developed based on Cunningham [36]. The latter section included six items, including "*Quizizz* helped me study the vocabulary sets independently.", "I could learn vocabulary autonomously on *Quizizz*.", and "I enjoyed learning vocabulary independently on *Quizizz*." The items were created by referring to Agustín-Llach and Alonso [59]. These two aspects received responses ranging from 1 to 5, with "1" indicating strong disagreement and "5" indicating strong agreement. The questionnaire items can be seen in Appendix A.

It is worth noting that the survey employed in this study adhered to the notions of learner training and autonomy raised by the two studies: Cunningham [36] and Agustín-Llach and Alonso [59]. The outcomes of the two studies were also used to construct survey items for learner training and autonomy using the Quizizz app. The survey adaptation focused on study ideas and findings rather than a direct adaptation of the two investigations.

To validate the questionnaire items, Cronbach's alpha was used to evaluate the internal consistency. After the data had been collected before the vocabulary learning started, the internal reliability was calculated. The results displayed high internal consistency for all the items on the use of *Quizizz* in supporting vocabulary learning ($\alpha = 0.878$) and on the use of *Quizizz* in developing learner autonomy in vocabulary learning ($\alpha = 0.899$). The reliability test results after the learning took place also indicated high internal consistency with $\alpha = 0.927$ and $\alpha = 0.892$, respectively.

The vocabulary test was the second instrument used, named Socrative Vocabulary Scores. Weekly vocabulary tests were administered to all students during the first 15 min of class using an online quiz application, i.e., *Socrative.com*. *Socrative* was selected to follow the university's paperless policy. The tests were designed to evaluate the results of each vocabulary list's autonomous vocabulary learning by the students. In one test, there were fifteen multiple-choice questions about word meanings, parts of speech, synonyms and antonyms, and sentence completion. The test lasted 10 min in class, with 5 min set aside for preparation. Students took the test by going to *Socrative.com* on their smartphones, and the teacher was able to monitor student progress from the classroom computer as well as display it on the screen projector. There were four vocabulary tests administered in total. The sample tests can be seen in Figure 2 below.

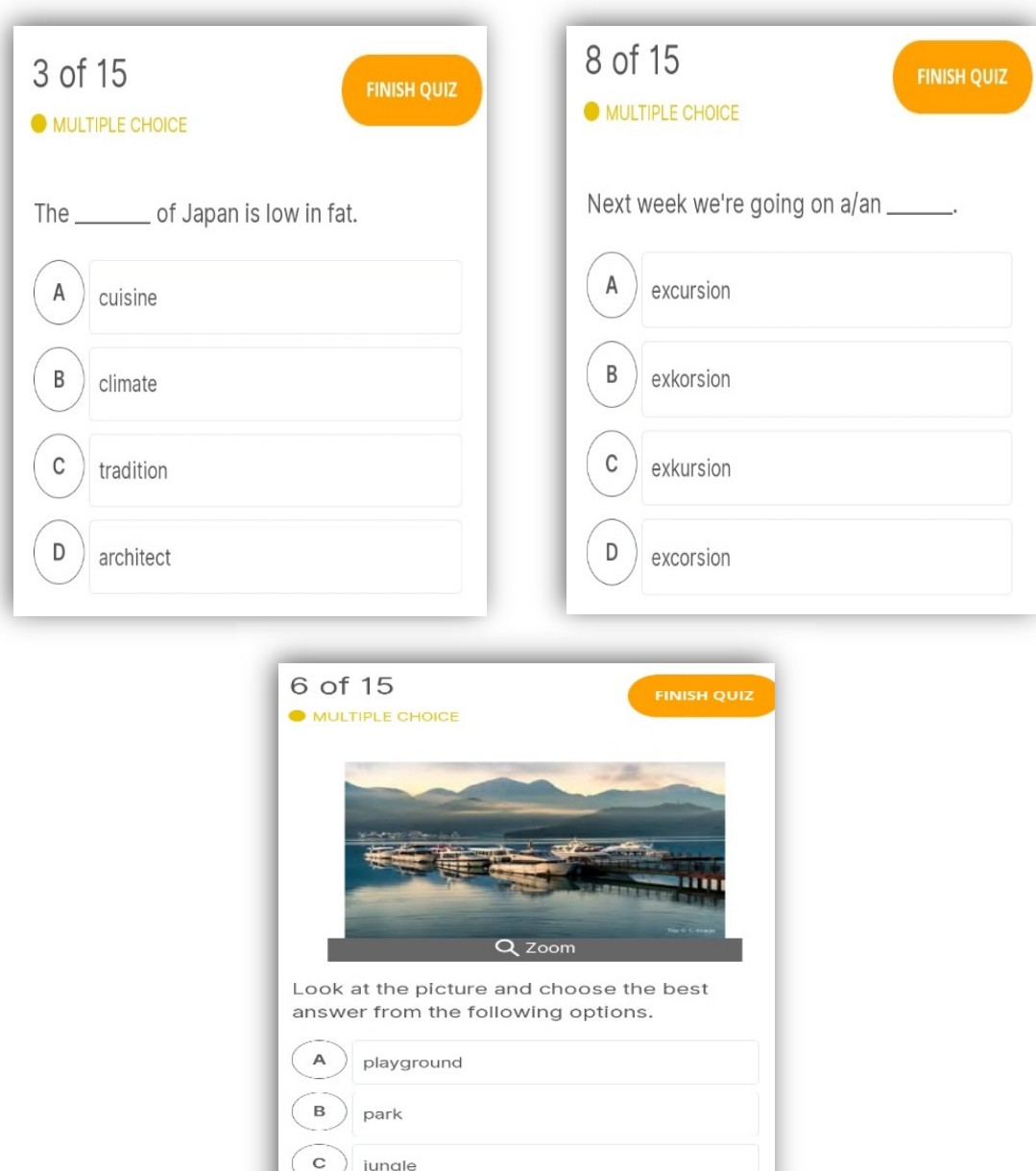

**Figure 2.** The vocabulary tests using Socrative.com.

The target vocabulary for the vocabulary tests was 200 academic English words at the CEFR levels A1-A2. The words were divided into four vocabulary lists, each of which contained 50 words. Students were required to complete the definitions and examples for each word in each list by consulting a dictionary at home. It was anticipated that students would develop their ability to learn vocabulary independently through this process. Some of the target words are presented in Table 1 while the whole list can be seen in Appendix B.

**Table 1.** The first 40 of 200 words that the students learned in 4 weeks.

| | | | |
|---|---|---|---|
| 1. Ability (n) | 11. Bowl (n) | 21. Text (v) | 31. Simple (adj) |
| 2. Permission (n) | 12. Stove (n) | 22. Publish (v) | 32. Instead (adv) |
| 3. Dark (adj) | 13. Visit (v) | 23. Married (adj) | 33. Stairs (n) |
| 4. Amazing (adj) | 14. Nearby (adj) | 24. Mistake (n) | 34. Complete (v) |
| 5. Familiar (adj) | 15. Jungle (n) | 25. Brilliant (adj) | 35. Blouse (n) |
| 6. Perhaps (adv) | 16. Homestay (n) | 26. Prefer (v) | 36. Match (n) |
| 7. Tasty (adj) | 17. Historix (adj) | 27. Through (prep) | 37. Cloudy (adj) |
| 8. Seat (n) | 18. Rural (adj) | 28. Afterwards (adv) | 38. Employ (v) |
| 9. Fry (v) | 19. Craft (n) | 29. Normal (adj) | 39. Agree (v) |
| 10. Grill (v) | 20. Access (v) | 30. Confident (adj) | 40. Awful (adj) |

Note. v = verb; n = noun; a = adverb; adj = adjective.

The last instrument was the students' weekly Quizizz practice scores. Researchers developed fifty multiple-choice questions using the words from each weekly vocabulary set. The scores ranged from 0 to 50. The practice questions were not the same as the test questions on the Socrative vocabulary tests in class. These data were gathered to see if students' practice scores corresponded to their in-class vocabulary tests (Socrative Vocabulary Scores) in each week. To obtain students' weekly Quizizz practice scores, researchers downloaded the reports provided by Quizizz for each week's practice. The reports were in Excel format, which gave details of students' names, IDs, and scores.

### 3.4. The Gamified Application: Quizizz.com

Description. *Quizizz.com* was employed in this study as an online gamification tool. *Quizizz* has garnered much attention for its gamified online quizzes. It accommodates both solo and group work and is suitable for in-class and take-home assignments that take the form of fill-in-the-blank, open-ended, or multiple-choice tests. Teachers can customize the exams to meet their teaching objectives and students' English proficiency levels. They can also add a timer, shuttle questions and answers, add memes and music, and display a leaderboard at the conclusion of each quiz. Students can see their live results and quiz evaluations following the test. *Quizizz* was superior to other applications in terms of both learning and assessment [60].

A gamified quiz application, such as *Quizizz.com*, is a type of game-based learning in which activities for lessons are created using game components [61–63]. It attempts to augment traditional teaching approaches, such as teacher-centered instruction and oriented assessments [64], as well as to increase learners' engagement in lessons and their learning results [65,66]. The quiz adds interest and intrigue to the courses. Learners are eager to voice their concerns about lessons and are motivated to complete challenging tasks. Additionally, gamified quizzes are critical for formative assessment [67]. It assists teachers in eliciting students' knowledge and tracing their emotional and cognitive characteristics through the use of game features such as scores, badges, rankings, achievement records, and leader boards [56,68]. As a result, teachers can conduct lessons more smoothly, and students may better develop the necessary abilities for academic training [63,69]. The illustration of the online quizzes on Quizizz.com is shown in Figure 3.

Rationales for selection. Earlier studies pointed out that the *Quizizz* quiz was more beneficial than other applications in assessments. It aids learners in becoming more attentive in responding to test items than the *Google Forms* and *Kahoot!* applications [60,63]. The quiz could lower learners' test anxiety and stimulate their performance in any exam [66,70]. Learners feel more relaxed in the test room, so they are able to concentrate on questions and deliver quick answers. This is in line with scholars' [62,63] findings of a positive correlation between learners' motivation and their test performance. Additionally, comparative to other popular applications such as Kahoot!, the amount of empirical research exploring the usage of *Quizizz* in ELT is still quite low. Examining a lesser-known gamification app

can help us expand our understanding of the extent to which gamification apps can assist English teaching and learning.

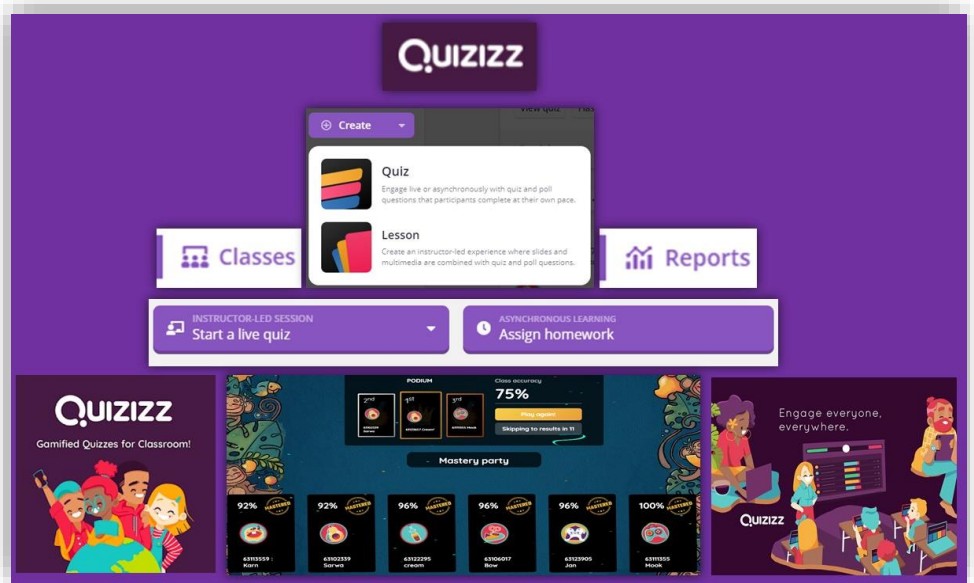

**Figure 3.** Illustration of the online quizzes on Quizizz.com.

*3.5. Research Procedures*

Stage 1—Preparation and Introduction. The 200 target words and four vocabulary lists were prepared prior to the start of the vocabulary learning. Then, four sets of quizzes on *Quizizz* were created to serve as a means of vocabulary training for students. The quizzes are depicted in Figure 4. The teacher provided a brief orientation of the course and vocabulary instruction to the class, including the given treatments in each group. Students were aware at this point that they would be required to study each vocabulary set weekly and would face vocabulary tests in class. The teacher informed the students that the four vocabulary sets had been incorporated into the course textbook. The target words were not discussed in class or taught. The first survey was distributed to students in the experimental group before the study began.

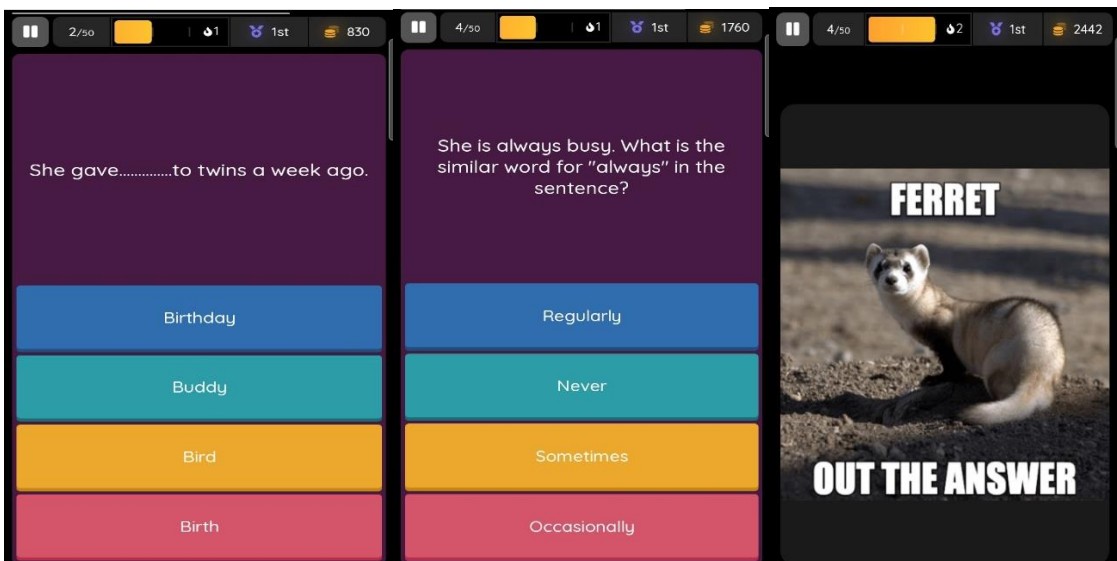

**Figure 4.** The Quizizz support along with one of the memes.

Stage 2—Implementation. Two different treatments were given to each group which were explained to the students during Stage 1.

Control group. Students were given one weekly vocabulary set containing 50 words at the CEFR levels A1–A2 to study independently at home. Students were required to complete the definitions and examples for each word in each vocabulary set by consulting a dictionary at home, which was briefly explained during Stage 1—Preparation and Introduction.

Experimental group. Aside from having one weekly vocabulary set that students have to study independently at home by consulting a dictionary at home, students were given *Quizizz* support in the forms of vocabulary exercises/practice. Students could access the *Quizizz* vocabulary exercises (practice) repeatedly and autonomously via a smartphone or computer from any location and at any time.

To ensure all the students proceeded as expected, during Stage 1—Preparation and Introduction, teachers emphasized that students had to study and complete the weekly vocabulary sets independently at home by consulting a dictionary for the control group or by consulting a dictionary and completing the weekly *Quizizz* vocabulary exercises/practice for the experimental group. *Quizizz* provided reports of students who completed the weekly *Quizizz* vocabulary exercises, which were used by researchers to monitor the progress of students in the experimental group, and these data were used for further investigation of the students in the experimental group.

In each weekly class, prior to the in-class vocabulary tests, students both groups submitted their completed vocabulary sets. Then, students in both groups took in-class vocabulary tests to assess the outcomes of their outside-of-classroom vocabulary learning. Furthermore, the investigation was extended for the experimental group by examining the performance of experimental group students on *Quizizz* practice and vocabulary tests to see if there was a link between completing the *Quizizz* exercises and students' vocabulary test scores.

Stage 3—Evaluation. Two types of data were analyzed: the students' vocabulary test scores and their responses to the survey questionnaire. To address the first research question, an independent *t*-test was used to determine whether there were significant differences in students' test scores between the experimental and control groups. Following that, data from the survey were analyzed using frequency and descriptive statistics. The mean was interpreted using three scales: Low level (1–2.4), Moderate level (2.5–3.4), and High level (3.5–5). Then, to assess the intervention's effectiveness, the survey results were compared before and after the learning process using paired-sample *t*-tests. Skewness and kurtosis values were used to determine the data's normality. According to George and Mallery [71], values between $-2$ and $+2$ indicate a normal distribution and the descriptive data from this data met these values, subjecting the data to parametric tests. The significance level was determined to be $p < 0.05$.

## 4. Results

### *4.1. The Intervention's Effects on Learning Outcomes*

Multiple independent *t*-tests were performed to examine the differences in vocabulary learning outcomes between the experimental and control groups. The differences were explored in the students' overall and weekly vocabulary test results in Socrative Vocabulary Scores. For the overall results, significant differences were not observed between students in both groups ($t$ (2, 98) = 1.920, $p = 0.06$) with a modest effect size: Cohen's d = (12.985 $-$ 12.435)/1.432245 = 0.384013. Out of 15, the average scores of the students in both groups were equal at 12, which was 80% of the total score, while the SD values were more than 1, indicating the existence of slightly unequal distributions among the students. Meanwhile, significant differences were noticed in the results of vocabulary tests 2 ($t$ (2, 98) = 3.229, $p = 0.002$) and 4 ($t$ (2, 98) = 3.465, $p = 0.001$) where the students in the experimental groups achieved higher scores ($M = 13.63$, $SD = 1.51$ and $M = 12.60$, $SD = 1.53$), yet no significant differences were noted in the results of vocabulary tests 1 and

2. Descriptively, as indicated in Figure 5, the control group's average scores in vocabulary test 1 were higher than the experimental group's, but lower in the other vocabulary tests and total test results.

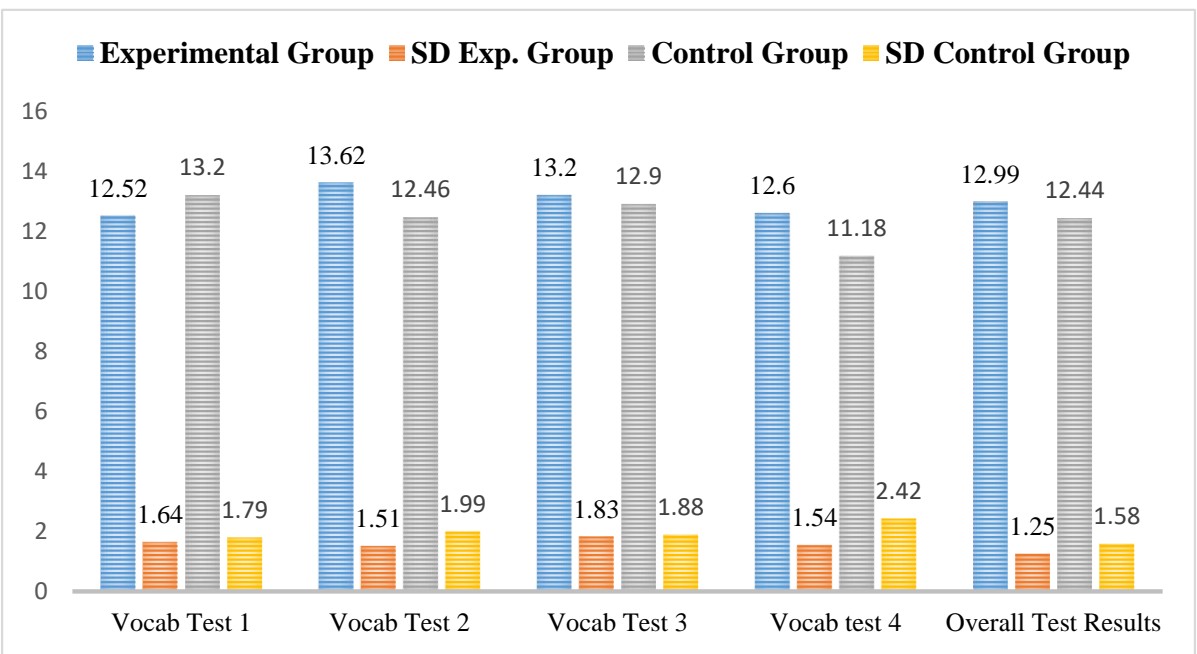

**Figure 5.** Descriptive comparison.

The investigation was furthered by examining the performance of experimental group students on *Quizizz* practices and Socrative Vocabulary tests. Table 2 shows significant and favorable relationships between students' *Quizizz* practice 1 ($M$ = 42.66, $SD$ = 5.11) and test 1 scores ($M$ = 12.52, $SD$ = 1.64), practice 2 ($M$ = 43.40, $SD$ = 7.31) and test 2 ($M$ = 13.62, $SD$ = 1.51), practice 3 ($M$ = 45.56, $SD$ = 7.31) and test 3 ($M$ = 13.2, $SD$ = 1.83), practice 4 ($M$ = 44.06, $SD$ = 7.82) and test 4 ($M$ = 12.6, $SD$ = 1.54), and overall practice results ($M$ = 42.49, $SD$ = 4.23) and overall Socrative vocabulary test results ($M$ = 12.99, $SD$ = 1.25). These results were significant because they demonstrated how the intervention in the experimental group affected students' vocabulary learning outcomes when compared to students in the control group who did not receive the intervention.

**Table 2.** Pearson correlations between students' Quizizz practice and Socrative vocabulary test scores.

| Vocabulary Test | | Quizizz Practice | | | | |
|---|---|---|---|---|---|---|
| | | 1 | 2 | 3 | 4 | Average |
| 1 | $r$ | 0.473 ** | 0.532 ** | 0.620 ** | 0.439 ** | 0.626 ** |
| | $p$ | 0.001 | 0.000 | 0.000 | 0.001 | 0.000 |
| 2 | $r$ | | 0.363 ** | 0.369 ** | 0.223 | 0.415 ** |
| | $p$ | | 0.01 | 0.008 | 0.119 | 0.003 |
| 3 | $r$ | | | 0.512 ** | 0.352 * | 0.517 ** |
| | $p$ | | | 0.000 | 0.012 | 0 |
| 4 | $r$ | | | | 0.314 * | 0.461 ** |
| | $p$ | | | | 0.026 | 0.001 |
| Average | $r$ | | | | | 0.650 ** |
| | $p$ | | | | | 0.000 |

* Correlation is significant at the 0.05 level (2-tailed); ** Correlation is significant at the 0.01 level (2-tailed).

### 4.2. Students' Perceptions

The survey questionnaires were distributed before and after the interventions. The results of paired-sampled *t*-tests displayed that significant changes occurred in students' perceptions of the application of gamification as a means for facilitating training ($t$ (1, 49) = 2.269, $p$ = 0.03) with a modest effect size: Cohen's d = (3.81 − 3.9425)/0.63649 = 0.208173. However, these significant results emphasized that students' positive perceptions decreased from before ($M$ = 3.94) to after the interventions ($M$ = 3.81). Nevertheless, their perceptions of the use of gamification for enhancing learner autonomy in vocabulary learning did not significantly change before and after the interventions ($t$ (1, 49) = 1.652, $p$ = 1.05) with a very small effect size: Cohen's d = (3.6067 − 3.73)/0.706969 = 0.174407. Furthermore, the descriptive statistics demonstrated that students rated *Quizizz* favorably in terms of learner training and autonomy, with all means larger than 3.5, as shown in Table 3.

**Table 3.** Descriptive results.

| Questionnaire Items | Before | | After | | |
|---|---|---|---|---|---|
| | **Mean** | **SD** | **Mean** | **SD** | **Level** |
| Quizizz as means for facilitating vocabulary training | | | | | |
| Quizizz.com from teacher really helped me learn the vocabulary sets more and better. | 4.10 | 0.79 | 3.96 | 0.81 | High |
| Quizizz.com from teacher really facilitated my vocabulary learning through practices. | 3.98 | 0.87 | 3.84 | 0.84 | High |
| Quizizz.com from teacher enabled me to practice on vocabulary exercises more. | 4.24 | 0.72 | 3.96 | 0.76 | High |
| I used Quizizz.com from teacher more than one time for my vocabulary practice every week. | 3.80 | 0.83 | 3.64 | 0.88 | High |
| I felt that I learned the vocabulary sets better using Quizizz.com from teacher. | 3.96 | 0.78 | 3.78 | 0.74 | High |
| My scores on vocabulary tests increased since using Quizizz from teacher. | 4.00 | 0.81 | 3.76 | 0.80 | High |
| Teacher should have created Quizizz since vocabulary test 1. | 3.76 | 0.94 | 3.86 | 0.81 | High |
| I liked the exercises on Quizizz.com | 3.70 | 0.91 | 3.68 | 0.87 | High |
| Average Scores | 3.94 | 0.61 | 3.81 | 0.66 | High |
| Quizizz as means for enhancing learner autonomy | | | | | |
| Quizizz.com helped me study the vocabulary sets independently. | 3.58 | 0.84 | 3.38 | 1.01 | High |
| I could learn vocabulary autonomously on Quizizz.com. | 3.76 | 0.89 | 3.58 | 0.84 | High |
| I enjoyed learning vocabulary independently on Quizizz.com. | 3.70 | 0.86 | 3.68 | 0.87 | High |
| I enjoyed learning vocabulary independently on Quizizz.com. | 3.62 | 0.88 | 3.64 | 0.83 | High |
| Quizizz.com supported my autonomous learning effectively. | 3.82 | 0.87 | 3.66 | 0.85 | High |
| I felt that I have become more independent in vocabulary learning since using Quizizz.com. | 3.90 | 0.84 | 3.70 | 0.89 | High |
| Average Scores | 3.73 | 0.70 | 3.61 | 0.71 | High |

## 5. Discussion

### 5.1. Findings

The main objective of this study was to examine the effects of gamified vocabulary learning through one of the gamification apps, *Quizizz*, in a quasi-experimental research design. It specifically explored students' perceptions of the use of a gamification app as a means for facilitating their vocabulary training outside the classroom and the effectiveness of enhancing learner autonomy in vocabulary learning. It also probed into how gamified vocabulary learning affects students' vocabulary learning outcomes. It is important to note that most previous gamified learning studies utilized the *Quizlet* app and positive results were obtained [7,15,35]. This app has a specific design for aiding vocabulary learning, which may contribute to the positive outcomes. Meanwhile, the present study used *Quizizz*, a gamification app not specifically designed for vocabulary learning, and is expected to shed light on how a non-specifically designed gamification app could be used to facilitate learner training and enhance learner autonomy in vocabulary learning. Based on the analysis results, there are two points worth discussing.

The first finding confirmed that, despite the high score results, there were no statistically significant differences in the total vocabulary scores of students in the experimental and control groups. On weekly assessments, substantial variations in students' vocabulary test scores showed only on tests 2 and 4, when experimental students outscored control group students, while they fared similarly on tests 1 and 3. Due to these contradictory findings, this study partially validates earlier research demonstrating the beneficial impacts of gamified vocabulary learning. Katemba and Sinuhaji [23] used the ESA (Engage, Study, Activate) method in conjunction with *Quizizz* to increase students' vocabulary knowledge and discovered that students in the experimental group achieved a higher level of proficiency and enjoyed vocabulary learning more than students in the control group who did not receive *Quizizz* assistance. Bal [55] conducted a 4-week study in which 60 students were divided into experimental and control groups to assess the association between MALL and vocabulary development. They discovered that the experimental group outperformed the control group somewhat; however, the difference in their scores was not statistically significant. According to Göksün and Gürsoy [56], the *Quizizz* application's limited visual feedback capability, question progression at an individual pace, and technological challenges could all be considered negatives. Therefore, the current study shall recognize the inconsistent results of gamifying vocabulary learning using *Quizizz*, especially for use outside of the class-hour. Nonetheless, the small number of available studies in this area may also be considered, and encouragement of integrating *Quizizz* into vocabulary learning may be tied to the students' backgrounds and learning contexts.

Moreover, the first finding indicated that using a gamification app not specifically designed for vocabulary learning, such as *Quizizz* in this study, gave a different result. The gamification integration would not result in different achievements between those who received *Quizizz* support and those who did not. Nonetheless, it is important to look at the designs of the previous studies too. For instance, Waluyo and Bakoko (2021) [34] integrated *Quizlet* for the study of 500 academic English words at the A2–B2 levels of the Common European Framework of Reference for Languages (CEFR), and Waluyo and Bucol (2021) [7] utilized *Quizlet* to support vocabulary learning of low-proficiency students in Thailand; both studies obtained positive results. Yet, the studies employed a single experimental design, in which they did not have a control group for result comparison. In contrast, the current study not only used a different gamification app but also employed a quasi-experimental research design that compared students' learning outcomes in experimental and control groups. Therefore, this study would not strongly state that the integration of *Quizizz* into vocabulary learning for learner training and autonomy would result in better outcomes compared to the one with traditional learning. Nonetheless, it lends support to early studies' suggestions about the importance of providing training exercises for students, particularly outside the classroom [31–33].

Additionally, because the first finding revealed no significant difference between the experimental and control groups, this study emphasizes that, while studying vocabulary using gamified apps outside of the classroom may seem like an interesting idea and may add some fun features to student learning, it is critical to keep in mind that the learning outcomes would likely be comparable to those studying vocabulary using a paper-based vocabulary list. This is also an outcome of the pedagogical context beyond the classroom. The majority of research on Quizizz focuses on the app's incorporation into classroom teaching and learning [24,54]. The benefits of gamification apps for vocabulary acquisition that Zou et al. [46] mention appear to be visible in in-class vocabulary learning. At this point, the second finding provides confirmation of how students felt and experienced vocabulary development during the research intervention.

The second finding disclosed that students' perceptions of the application of gamification as a means for facilitating training significantly decreased from before to after the interventions. In contrast, their perceptions of the use of gamification for enhancing learner autonomy in vocabulary learning did not significantly change after the interventions. It is critical to note, however, that despite these declines and non-significant variations, students' opinions of *Quizizz* use remained high (means > 3.5). As a result of assuming that students practiced vocabulary independently prior to the study, significant differences between before and after the interventions were not observed, but students' perceptions of the interventions remained positive due to their understanding of the interventions' benefits to their vocabulary learning [41]. Chan [45] asserted that EFL students possess a fundamental understanding of the value of independent learning and the capacity to practice it, with further development contingent on how teachers build the learning environment for students. Once a student assumes ownership of his or her own learning, he or she will be able to establish learning objectives, manage appropriate learning techniques, and engage in autonomous learning [42]. Students who receive assistance with gamification are encouraged to handle their vocabulary practices independently at home [47–49]. Simultaneously, the game aspects added to a gamification application boost students' enjoyment of learning, motivating them to practice with the app more frequently [50].

In comparison to non-English teaching studies, the current study's findings support the use of gamified online quizzes to help students practice their homework or exercises in preparation for in-class assessments. A link was discovered between learning achievement as measured by final exam marks and participation in online quizzes, as well as the learning gained from completing these quizzes [38]. The study also found that the amount of time spent, the number of attempts, and the quiz scores all had a positive effect on student performance on in-class tests [39].

### 5.2. Implication and Limitation

The findings have implications for pedagogy and research. First, as various experts have indicated, gamification apps can be integrated into vocabulary learning to facilitate vocabulary study and foster the growth of learner autonomy [14–16,36]. However, utilizing a non-specifically built app for vocabulary learning may not produce better results than those who do not receive gamification support, even if learning outcomes remain good. Second, the findings suggest additional implementation of quasi-experimental research exploring the possibility of a non-specifically built app for learner instruction and autonomy.

Nonetheless, this study acknowledges several limitations. It did not investigate students' learning experiences qualitatively, which prevented it from getting individual insights. We intended to include interview data during the course of the study; however, this was canceled due to the emergence of the COVID-19 pandemic. The findings should be interpreted in the context of a quasi-experimental research design involving experimental and control groups, meaning that empirical studies utilizing different research designs may or may not obtain similar results. The duration of the intervention was also short, which may have had an impact on the findings because the students may have needed some time to adjust to the *Quizizz* practice.

## 6. Conclusions

Incorporating a gamification app for facilitating vocabulary study and supporting the development of learner autonomy outside of the classroom may alleviate the difficulties posed by restricted class time for vocabulary learning. As a response, this study explored how the integration of a gamification app that was not specifically designed for vocabulary learning, namely *Quizizz*, affected the outcomes and perceptions of students' vocabulary learning. In conclusion, the gamification integration would not result in distinct accomplishments for those who received support from *Quizizz* and those who did not. While studying vocabulary using gamified apps outside of the classroom may seem like an interesting idea and may add some fun elements to student learning, the learning outcomes are likely to be comparable to those of students studying vocabulary using a paper-based vocabulary list outside of the classroom. Students' perceptions of gamification as a means for facilitating training decreased significantly from before to after the interventions, whereas their perceptions of gamification as a means for enhancing learner autonomy in vocabulary learning did not change significantly after the interventions. Despite these declines and non-significant variations, students' views of *Quizizz* use remained high (means > 3.5) and positive. This research issue still has a scarcity of empirical investigations that use an experimental research design. As a result, it is suggested that future studies employ an experimental research design. Furthermore, qualitative data may provide a more complete picture of students' learning experiences with gamification apps.

**Author Contributions:** Conceptualization, B.P. and B.W.; methodology, B.P.; validation, B.P.; formal analysis, B.W.; resources, B.P.; writing—original draft preparation, B.P. and B.W.; writing—review and editing, B.P. and B.W.; visualization, B.P. and B.W.; supervision, B.W.; project administration, B.P. and B.W. All authors have read and agreed to the published version of the manuscript.

**Funding:** This research received no external funding.

**Institutional Review Board Statement:** This study was approved by the ethics committee in human research, Walailak University, Thailand (WUEC-21-237-01).

**Informed Consent Statement:** Participation was voluntary and had no impact on grades or academic performance. The researchers used identification numbers to identify each student in this study.

**Data Availability Statement:** Data are available upon request.

**Conflicts of Interest:** The authors declare no conflict of interest.

## Appendix A. The Questionnaire Items

| Questionnaire Items | Strongly Disagree | Disagree | Neutral | Agree | Strongly Agree |
|---|---|---|---|---|---|
| Quizizz as means for facilitating vocabulary training | | | | | |
| Quizizz.com from teacher really helped me learn the vocabulary sets more and better. | | | | | |
| Quizizz.com from teacher really facilitated my vocabulary learning through practices. | | | | | |
| Quizizz.com from teacher enabled me to practice on vocabulary exercises more. | | | | | |
| I used Quizizz.com from teacher more than one time for my vocabulary practice every week. | | | | | |
| I felt that I learned the vocabulary sets better using Quizizz.com from teacher. | | | | | |
| My scores on vocabulary tests increased since using Quizizz from teacher. | | | | | |

| Questionnaire Items | Strongly Disagree | Disagree | Neutral | Agree | Strongly Agree |
|---|---|---|---|---|---|
| Teacher should have created Quizizz since vocabulary test 1. | | | | | |
| I liked the exercises on Quizizz.com | | | | | |
| Quizizz as means for enhancing learner autonomy | | | | | |
| Quizizz.com helped me study the vocabulary sets independently. | | | | | |
| I could learn vocabulary autonomously on Quizizz.com. | | | | | |
| I enjoyed learning vocabulary independently on Quizizz.com. | | | | | |
| I enjoyed learning vocabulary independently on Quizizz.com. | | | | | |
| Quizizz.com supported my autonomous learning effectively. | | | | | |
| I felt that I have become more independent in vocabulary learning since using Quizizz.com. | | | | | |

**Appendix B. The 200 Words that the Students Learned in 4 Weeks**

| | | | |
|---|---|---|---|
| 1. restaurant (n) | 2. holiday (n) | 3. access (v) | 4. storm (n) |
| 5. homemade (adj) | 6. visit (v) | 7. click (v) | 8. revise (v) |
| 9. lasagna (n) | 10. magical (adj) | 11. notice (n) | 12. dictionary (n) |
| 13. ability (n) | 14. experience (n) | 15. tonight (n) | 16. playground (n) |
| 17. permission (n) | 18. non-stop (adj) | 19. paragraph (n) | 20. classmate (n) |
| 21. probability (n) | 22. square (n) | 23. upstairs (n) | 24. assistant (n) |
| 25. request (n) | 26. wander (v) | 27. text (v) | 28. wear (v) |
| 29. suggestion (n) | 30. souvenir (n) | 31. publish (v) | 32. timetable (n) |
| 33. necessary (n) | 34. jacket (n) | 35. adult (n) | 36. essay (n) |
| 37. match (n) | 38. kebab (n) | 39. title (n) | 40. bandage (n) |
| 41. article (n) | 42. lentil (n) | 43. continue (v) | 44. background (n) |
| 45. dark (adj) | 46. eggplant (n) | 47. pack (v) | 48. licence (n) |
| 49. amazing (adj) | 50. tower (n) | 51. repair (v) | 52. friendship (n) |
| 53. familiar (adj) | 54. cross (v) | 55. arrive (v) | 56. cousin (n) |
| 57. novel (n) | 58. bridge (n) | 59. advice (n) | 60. beard (n) |
| 61. address (n) | 62. mosque (n) | 63. article (n) | 64. sausage (n) |
| 65. without (prep) | 66. scarf (n) | 67. sheet (n) | 68. librarian (n) |
| 69. among (prep) | 70. vendor (n) | 71. kite (n) | 72. through (prep) |
| 73. perhaps (adv) | 74. although (conj) | 75. private (adj) | 76. rest (n) |
| 77. example (n) | 78. landmark (n) | 79. treat (v) | 80. polite (adj) |
| 81. polite (adj) | 82. neighborhood (n) | 83. advertisement (n) | 84. afterwards (adv) |

| | | | |
|---|---|---|---|
| 85. dessert (n) | 86. pier (n) | 87. apartment (n) | 88. turn (vj) |
| 89. tasty (adj) | 90. nearby (adj) | 91. grow (v) | 92. leather (n) |
| 93. fresh (adj) | 94. marine (adj) | 95. furniture (n) | 96. appointment (n) |
| 97. dish (n) | 98. jungle (n) | 99. housework (n) | 100. normal (adj) |
| 101. waiter (n) | 102. cuisine (n) | 103. level (n) | 104. envelope (n) |
| 105. cafeteria (n) | 106. sunbathe (v) | 107. biscuit (v) | 108. nervous (adj) |
| 109. canteen (n) | 110. snorkel (v) | 111. conversation (n) | 112. confident (adj) |
| 113. soup (n) | 114. flip-flops (n) | 115. collect (v) | 116. simple (adj) |
| 117. atmosphere (n) | 118. diving (n) | 119. married (adj) | 120. instead (advj) |
| 121. seat (n) | 122. camping (n) | 123. develop (v) | 124. along (prep) |
| 125. option (n) | 126. excursion (n) | 127. trend (n) | 128. stairs (n) |
| 129. beverage (n) | 130. villa (n) | 131. chance (n) | 132. complete (v) |
| 133. reservation (n) | 134. brochure (n) | 135. mistake (n) | 136. blouse (n) |
| 137. ingredient (n) | 138. inn (n) | 139. mind (v) | 140. electric (adj) |
| 141. recommend (v) | 142. homestay (n) | 143. trouble (n) | 144. laptop (n) |
| 145. bitter (adj) | 146. lobby (n) | 147. brilliant (adj) | 148. match (n) |
| 149. salty (adj) | 150. budget (n) | 151. modern (adj) | 152. headache (n) |
| 153. sour (adj) | 154. facility (n) | 155. manage (v) | 156. cloudy (adj) |
| 157. tender (adj) | 158. balcony (n) | 159. prefer (v) | 160. spare (adj) |
| 161. greasy (adj) | 162. spacious (adj) | 163. success (n) | 164. during (prep) |
| 165. fry (v) | 166. buffet (n) | 167. win (v) | 168. recent (adj) |
| 169. stir (v) | 170. fill in (phr v) | 171. lend (v) | 172. regular (adj) |
| 173. roast (v) | 174. getaway (n) | 175. aim (n) | 176. suddenly (adv) |
| 177. grill (v) | 178. hang out (phr v) | 179. lamp (n) | 180. necklace (n) |
| 181. chop (v) | 182. attraction (n) | 183. goal (n) | 184. employ (v) |
| 185. bowl (n) | 186. rural (adj) | 187. ticket (n) | 188. comb (n) |
| 189. stove (n) | 190. gallery (n) | 191. send (v) | 192. receipt (adj) |
| 193. fork (n) | 194. craft (n) | 195. screen (n) | 196. agree (v) |
| 197. spoon (n) | 198. historic (adj) | 199. reason (n) | 200. awful (adj) |

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
