# Peer review of "The Pedagogical Use of Gamification in English Vocabulary Training and Learning in Higher Education"

_education, doi:10.3390/educsci13010024_

Round 1

Reviewer 1 Report

This is an interesting paper about gamification in English vocabulary learning through Quizizz. However, I have raised some issues and comments for the author/s:

 1.    Some claims need to be proven, for example ‘The number studies investigating the integration of other gamification apps into vocabulary learning outside the classroom are still scarce’ (page 2, 87-88). Which ones? There is today a good number of studies about the adoption of different game-based mobiles apps to learn different language skills (Kahoot, Quizalize, etc). The author/s should clearly mention them as this is related with the research need. There are several studies and reviews about this issue, for example:

- Chen, C. M., Liu, H., & Huang, H. B. (2019). Effects of a mobile game-based English vocabulary learning app on learners’ perceptions and learning performance: A case study of Taiwanese EFL learners. ReCALL, 31(2), 170-188.

-Xu, Z., Chen, Z., Eutsler, L., Geng, Z., & Kogut, A. (2020). A scoping review of digital game-based technology on English language learning. Educational Technology Research and Development, 68(3), 877-904.

  2.  Some statements need to be better supported. For example, ‘Besides, most of gamification apps are commonly not specifically created for teaching English or specific English skills’ (page 2, 88-89). Several studies have been recently published about the use of game-based apps which are specifically designed for language learning, for example:

-Gunter, G. A., Campbell, L. O., Braga, J., Racilan, M., & Souza, V. V. S. (2016). Language learning apps or games: an investigation utilizing the RETAIN model. Revista Brasileira de Linguística Aplicada, 16, 209-235.

- Kohnke, L., Zhang, R., & Zou, D. (2019). Using mobile vocabulary learning apps as aids to knowledge retention: Business vocabulary acquisition. Journal of Asia TEFL, 16(2), 683.

- Belda-Medina, J., & Calvo-Ferrer, J. R. (2022). Preservice Teachers’ Knowledge and Attitudes toward Digital-Game-Based Language Learning. Education Sciences, 12(3), 182.

 3.    Method: The intervention should be better explained, this is the major challenge in my opinion. According to the author/s ‘The experimental groups received the gamified learning intervention during their vocabulary learning while the other group did not.’ (page 6, 285-286). So, what did the CG participants actually do? What kind of treatment did they receive?  It seems that the EG received clear instructions (Quizizz) as opposed to the CG, for example ‘For the control group, students were assigned to study each vocabulary set independently at home each week using any method they desired’ (page 11, 465-467). This is rather vague, what method/s did they use?

4.    Method: And how did the EG learn the vocabulary using Quizizz? How did the author/s know the participants were actually using it (frequency, control, etc)?

5.    How were the research participants assigned to the EG and CG? Which was the selection criteria?

 6.    Instruments: The questionnaire should be included in an Appendix to avoid last paragraph on page 7 (lines 337-358).

 7.    Method: Why did the author/s use Socrative for the English vocabulary test? Did all participants use it? This is another game-based app.

 8.    Some conclusions seem not to be aligned with the research findings. For example, ‘Gamification apps not only allow students to expand their vocabulary learning beyond the classroom, but they may also transform the learning experience’ (page 16, 643-644) versus ‘there were no statistically significant differences in the total vocabulary scores of students in the experimental and control groups’ (page 14, 543-544)

9. The limitations should make clear reference to the type/s of methods employed with the EG and the CG and their impact on the learning outcomes.

Reviewer 3 Report

This is an interesting, well-developed and very well-written paper.  The aim is clear and the title is informative and the conceptualisation is thoroughly presented. The variables were defined and measured appropriately and the study methods are valid and reliable.  The results are clearly presented, expand previous knowledge, and are quite useful for those who are willing to introduce gamification activities into their teaching practices. Congratulations to the author(s) on a well-achieved work.

Round 2

Reviewer 1 Report

Dear author/s,

Thanks for your response and comments. I still believe the literature review needs to be reinforced as suggested in my previous review. Some statements lack specificity. Quizziz is not a language-oriented app and your research is based on it. In fact, there are other similar apps such as Quizlet, Quizalize, Kahoot, etc which have been widely studied and even compared to Quizziz, and there are more specific language-related gamified apps for vocabulary learning as previously suggested.

However, the major issue is related with the treatment. The intervention with the control group (CG) is not clearly defined. I think the results can be biased based on the fact that more attention and specific rules were provided to the EG. From your description, it seems that the CG could use 'any method' they wanted. Which ones? How did they select it/them? How did you control it?

In your own words, 'students were required to independently study each vocabulary set at home once per week using any method they wanted '. This is rather vague as you were allegedly comparing the effect of using a (non specifically language-related) gamified app for vocabulary learning with 'any method' the students wanted to use. In such case, how could the effect (vocabulary achievement) be properly measured if  we do not know the methods (if any) used by the CG? What selection criteria did the CG follow to select the 'other/any method'? What if they (CG) used different methods? And how could you monitor and control (time, tool, method, etc) the intervention without previously providing clear indications to the CG?

There seems to be a clear disadvantge between both groups in the intervention. In your response you further explain the treatment with the EG based on Quizizz (10 lines) but not with the CG (5 lines). As indicated in my previous response, the major concern is related with the treatment (method, time of exposure, monitoring, etc) of the CG.

As it stands now, it seems that  your expectations may have influenced the research design (Quizizz) and data collection process, also known as 'researcher bias'. I expected more specific details about the CG as this is the key issue in your research (control missing).

Thanks.

Reviewer 2 Report

It is a pleasure to see that the manuscript has improved a lot since the previous version. I find that you have addressed all my concerns in my previous review. Including a section “The Study” right before the Methods section improves the structure of the background. The Quizizz practice data is now included as a variable under “Instruments”, r. 417. And, most importantly, the conclusions are now more consistent with the reported results. In cases where you decide to go against my recommendations, you provide sound arguments for your decisions. After all, we do not always have to do as the reviewers say!

The manuscript may still be improved, but it will not require any major changes. Here are some suggestions:

p. 393: You could call this variable Socrative Vocabulary scores throughout the text, to make it clear that it is a different variable than the Quizizz Practice scores. Both are vocabulary tests, and by labelling them clearly, readers will not be confused as to what vocabulary test you are referring to.

r. 417: Can you provide more detail, for example how the Quizizz practice scores were obtained? I assume that you implemented the practice tests in the Quizizz application and that the questions were different (both in content and in format) from the questions they received in the Socrative application? The Socrative tests are carefully described, so the Quizizz practice tests deserve some more attention.

r. 510: Results section

The questionnaire data are well described in this paper, with both descriptive statistics (mean, sd, skewness etc) and alpha. However, the descriptive statistics for the vocabulary tests (Quizizz practice and Socrative) are not that carefully reported. As I wrote in my first review, it is understandable that you may not have access to all descriptive statistics from Quizizz and Socrative, but you should at least point this out to the readers. Normally, I would expect to find means and distributions (sd, skewness etc.) and reliability estimates for all variables in the study. One reason why this is important is that correlations (which you report in Table 2) are strongly influenced by the distributions of the correlated variables. Apparently, you do have access to standard deviations from Socrative, since you write “SD values were more than 1, indicating the existence of slightly unequal distributions among the students.” (r. 518). You may want to consider reporting means and standard deviations in a table, in which you could also include distributional descriptives for the Quizizz practice data.

The figure called “Chart 1” (by the way, at least in APA style it is customary to refer to this as a Figure, with appropriate number and title) is not very informative and could easily be improved. One way is to display score distributions by using box-and-whisker plots instead of just bars showing mean values. Another option for making Chart 1 more interesting is to add error bars indicating 95% confidence intervals for the means. Then, readers will be able to visually confirm any significant (or non-significant) differences just by looking at the chart.   
